# What can make things better for parents when babies need abdominal surgery in their first year of life? A qualitative interview study in the UK

Lisa Hinton,[1,2] Louise Locock,[3] Anna-May Long,[4] Marian Knight[4]

¹Nuffield Department of Primary Care Health Sciences, University of Oxford, Oxford, UK
²NIHR Oxford Biomedical Research Centre, Oxford, UK
³Health Service Research, University of Aberdeen, Aberdeen, UK
⁴Nuffield Department of Population Health, National Perinatal Epidemiology Unit, University of Oxford, Oxford, UK

**Correspondence to**
Dr Lisa Hinton;
lisa.hinton@phc.ox.ac.uk

## ABSTRACT

**Objectives** To understand the experiences of parents of infants who required surgery early in life. To identify messages and training needs for the extended clinical teams caring for these families—including paediatric surgeons, neonatologists, nurses, obstetricians, midwives and sonographers.

**Setting** UK-wide interview study, including England, Wales and Scotland.

**Participants** In-depth interviews were conducted with 44 parents who had a baby who underwent early abdominal surgery. Conditions included those diagnosed antenatally (eg, exomphalos, gastroschisis, congenital diaphragmatic hernia) or those which were detected postnatally (eg, Hirschsprung's disease, necrotising enterocolitis). Interviews were video and audio recorded and analysed using a modified grounded-theory approach.

**Results** While some parents reported experiencing excellent communication and felt they were listened to and involved by the care team, this was not always the case. Dealing with large, complex medical and surgical teams could result in conflicting messages, uncertainty and distress. Parents wanted information but also described being overwhelmed and wanting to distance themselves to maintain hope. Information and support from other parents in hospital and online groups were highly valued. Of particular concern was support when going home and caring for their baby after discharge; an open access policy for readmission offered a helpful safety net.

**Conclusions** Listening to the experience of parents provides rich data to enhance clinical understandings on how to improve information and communication with parents, and ameliorate the deep and lasting distress and anxiety that some parents feel when their infants face early surgery. We suggest that the writings of Bourdieu could have resonance in interpreting the experiences of parents as they enter the world of highly technical neonatal medicine and surgery and the knowledge of the professionals who work in these environments.

## INTRODUCTION

The experiences of parents of infants who undergo abdominal surgery in the first year of life are rarely recorded. Even after discharge from hospital, these infants may require multiple outpatient visits to different

### Strengths and limitations of this study

► This is a UK-based study that sought to explore the experiences of parents who have an infant requiring abdominal surgery in the first year of life.
► Interview studies in this field are rare and our study provides rich insights into the emotional and practical impacts of these experiences.
► While the interviews could not cover all possible conditions, they provide insights that may be generalisable across many different surgical procedures/medical conditions.
► Interviews were conducted at varying distances from the event, so recall was varied. Parents described an intense and often protracted period of their lives, so some details may have been omitted or compressed in their accounts.

specialist clinics and their ongoing care requirements can have significant effects on the quality of life of parents and existing children, as well as economic effects through lost work days.

Many of these infants will require surgery because of a gastrointestinal congenital anomaly, such as exomphalos, gastroschisis or Hirschsprung's disease. These conditions are under-researched[1]; understanding of them is poor and interventions are rarely based on robust evidence. Routine sources of information are limited.[2] Outcomes for infants with rare anomalies are improving[3 4] and so the need for rigorous research into surgical, clinical and long-term outcomes is paramount.[2] Alongside this, understanding the impact of these diagnoses and surgical treatments on parents and families is central to their long-term recovery.

This study was conducted as an adjunct to the British Association of Paediatric Surgeons Congenital Anomaly Surveillance System and was focused on parents' experiences of abdominal surgery. There is little research on parent experiences of conditions that

require early abdominal surgery. However, we can supplement this limited evidence base by drawing on research into parent experiences of congenital heart disease and surgery and more generally research about parents' experiences of neonatal intensive care units. The literature here points to the practical and psychological impact on families[5 6] and the importance of good communication between parents and staff.[7] Parents of children with congenital heart disease are at high risk for mental health morbidity, experiencing stress, depression, anxiety and traumatic stress responses[8–12] and the cost implications are profound.[13] Parents experience difficulties with bonding and breast feeding while their infants are in hospital[8 14–16] and their infants' complex needs after discharge.[17]

Mothers of infants in neonatal intensive care face emotional challenges as they seek to find a role for themselves in medically complex environments with perilously ill infants.[18] Wilkinson[19] and others have explored the contentious ethical issues facing those who care for critically ill infants and the extent to which parents should be involved in decision-making. Little is known about the relationship between quality of mother–child interaction and child development after major neonatal surgery or intensive care, but the parental mental health and familial environment are thought to be vital to the recovery and mental well-being of the infant.[20–22]

### Theoretical framework

One approach to thinking about these experiences could be to draw on the writings of the French sociologist Pierre Bourdieu whose work focused on examining experiences of social class and the production of knowledge and power.[23 24] One of his key concepts was the notion of 'habitus', whereby individuals share their history and environment with others who have similar positions, producing a 'stylistic affinity' and where 'social identity is defined and asserted through difference'. Habitus is the way one unconsciously acts, interacts and behaves within the social world in a 'taken for granted' manner, according to socialised norms, traditions and unwritten rules of particular groups.[25] Bourdieu identifies various forms of 'capital', economic, social and cultural resources that establish social standing within a particular habitus or social setting. Power is derived from configurations of these types of capital[26] within a given field (in this case neonatal care and surgery). Although his writings focused on class lifestyles and social identity, these ideas could have resonance in interpreting the experiences of parents as they enter the world of highly technical neonatal medicine and surgery and the knowledge of the professionals who work in these environments.[27]

The aim of this qualitative study was to: explore parents' experiences and perspectives of having a baby who needs early abdominal surgery; identify the questions and problems that matter to parents during and after their pregnancy and infant's surgery and identify the long-term impact on parents and families. In this article,

we report on the period immediately before and after surgery, and suggest actions that can be implemented to rapidly improve parents' experiences. Findings on the longer term impacts of these experiences will be reported elsewhere.

## METHODS

We conducted interviews with 44 parents, all of whom had an infant who needed abdominal surgery in the first year of life. Two parents did not return their copyright form so their data are not included in our analysis.

### Participants

Parents were recruited from England, Scotland and Wales. We interviewed parents whose infant had required surgery in their first year of life. We were keen to capture long-term perspectives on the experiences, as well as recent ones. So, we sought parents whose infant or child was now a wide variety of differing ages. Therefore, participants included those for whom the experience was very recent through to parents whose son was 25 years old. Most parents' infants were still living, but we interviewed the parents of one boy who died aged 7 months. We interviewed some mothers individually and 11 couples (mother and father). Conditions that their infants had been diagnosed with included exomphalos, gastroschisis, congenital diaphragmatic hernia, Hirschsprung's disease, inguinal hernia, atresia (jejunal and oesophageal) and necrotising enterocolitis (see table 1). Some conditions were diagnosed antenatally; others developed after their baby was born. Some conditions were congenital; others developed as a result of premature birth or spontaneously (see table 1). All infants had their surgery in the first year of life, although some went on to have multiple operations. Several spent extended periods of time in neonatal intensive care units (see table 1).

### Interviews

Parents were interviewed either individually or in pairs, depending on preference. We interviewed 11 fathers, but none chose to be interviewed on their own. Recruitment was through support groups, paediatric surgeons, neonatal nurses, other specialists and word of mouth. We aimed for a maximum variation sample and continued interviews until thematic saturation was reached. All interviews were conducted by LH, a social scientist, in participants' own homes, or in a venue of their choice using a semistructured narrative approach.[28] The interviews were in two parts, beginning with an invitation to offer an unstructured narrative, prompted by an open-ended question at the start of the interview. This was followed up by a semistructured component of the interview with prompts to follow-up issues raised in the narrative and to explore themes suggested by the literature and the advisory panel and a group of parent advisors. The interview schedule was developed by co-authors in consultation with the study advisory panel and parent

**Table 1** Participants

| Condition | Identifier | Parent Mother/father | Sex of child | Child's age at interview | Intensive or high dependency care admission |
|---|---|---|---|---|---|
| Exomphalos* | 1 | M | m | 18 months | Yes |
| | 2 | M | f | 6 years | Yes |
| | 3 | M | f | 8 years | Yes |
| | 5 | M and F | f | 8 years | Yes |
| | 13 | M and F | m | 25 years | No |
| | 16/17 | M and F | m | 13 years | Yes |
| | 19 | M | f | 16 months | Yes |
| | 41/42 | M and F | m | Died age 7 months | Yes |
| Necrotising enterocolitis | 8 | M | m | 9 months | Yes |
| | 9 | M | m | 11 months | Yes |
| | 23 | M | m | 6 months | Yes |
| Hernia | 26/27 | M and F | f | 9 months | No |
| Congenital diaphragmatic hernia* | 11 | M and F | m | 9 months | Yes |
| | 15 | M | m | 5 years | Yes |
| | 24 | M | m | 9 years | Yes |
| Hirschsprung's disease* | 18 | M | f | 1 year | No |
| | 20/21 | M and F | m | 4 months | Yes |
| | 22 | M | m | 16 months | No |
| | 39/40 | M and F | m | 5 years old | No |
| | 43/44 | M and F | m | 5 years old | No |
| Gastroschisis* | 30 | M | m | 7 ½ months | Yes |
| | 31 | M | m | 4 years | Yes |
| | 32 | M | f | 3 years | Yes |
| | 33/34 | M and F | m | 3 months | Yes |
| | 36 | M | f | 19 months | Yes |
| | 37 | M | f | 6 years | Yes |
| Short bowel | 35 | M | f | 7 years | Yes |
| Jejunal atresia* | 25 | M | f | 4 months | Yes |
| Oesophageal atresia with tracheo-oesophageal fistula* | 38 | | m | 19 months | Yes |
| Undiagnosed | 14 | | m | 6 months | |

*Congenital diagnosis.

advisors. All interviews were audio recorded, and, where consent was given, video recorded to facilitate dissemination on the patient experience website, Healthtalk.org. Parents were offered a two-stage consent and copyright approval process. They signed a consent form before the interview started and were subsequently sent a copy of their transcript to approve before signing a copyright form agreeing to their data being included in the analysis and excerpts from their interviews included in dissemination, including peer-review articles, education materials and on the Healthtalk.org website. Those who wanted to remain anonymous could choose to have only audio or written-only excerpts of their interviews included, and a pseudonym. Interviews lasted from between 75 min and 3 and a half hours. All interviews were fully transcribed. We undertook analysis of the transcribed interviews using an interpretive approach to thematic analysis.[29 30] Interview transcripts, not videos, were coded by LH with support from NVIVO analysis software, using a framework which was developed iteratively and reflected both anticipated and emergent themes. Coding reports were then analysed separately by LH and LL using a modified grounded theory approach, incorporating constant comparison and exploration of deviant cases, allowing the data to be grouped into themes and all cases to be examined to ensure all the manifestations of each theme

were accounted for and compared[31] and we were satisfied we had reached data saturation with no new constructs emerging.[30] Any differences in interpretation were then discussed. MK and AML reviewed summarised findings and contributed a further clinical layer of analysis.

## Patient and public involvement

A parent advisory panel was established at the inception of the study and met annually. Parents with lived experiences of having an infant who required neonatal surgery were involved in the face-to-face meetings or via email. Parents contributed to the design of the sample and interview schedule, helped with recruitment and provided feedback on initial analysis and the final documents published on the Healthtalk.org website. Parents were invited to a launch event for the website.

All participants gave informed consent before taking part and have given written consent to their interview data being included in publications.

## RESULTS

Some parents discovered that there was a problem with their baby during antenatal scans, others not until after their baby had been born. Regardless of when the diagnosis came, parents of a baby who needs abdominal surgery faced many challenges over the ensuing months and years.

The care pathways for these infants were often long and complex. Setbacks were common, and at no point could clinical staff give parents any guarantees. The practical and emotional uncertainties were, therefore, huge. Living with uncertainty and worry was something parents had to get used to, at every stage. Many parents described a lasting distress and anxiety.

> *It was always a waiting game and tricky because you want to know the answer, 'When is my baby going to come home?'* (ID02, mother, daughter with gastroschisis)

> *We just don't know what road he is going to take.* (ID20, mother, son with Hirschsprung's disease)

> *My anxiety levels have been awful really it's been hard for everybody because I've been up and down emotionally.* (ID23, mother, son with NEC)

Large and complex teams of staff across many different specialities were involved in looking after these infants over weeks, months and years. Our interviews highlighted some key steps that could improve the experience for parents. These do not so much focus on the surgery itself, but on the care and support parents are given before and after.

i.    Communication.
ii.   Managing information and information overload.
iii.  Encouraging parents to seek out others (online, hospital groups).
iv.   Help in finding a role
v.    Preparing for going home.
vi.   Open access policy.

## Communication

Communication with the various health professionals looking after their baby was central to parents' experiences. These professionals spanned many disciplines— neonatologists, surgeons, but also paediatricians, intensive care nurses, specialist stoma nurses and other allied health professionals further along, such as physiotherapists or nutritionists.

The babies were often very sick with a rare condition and receiving complex medical and surgical treatments. Parents often felt isolated and struggled to find information about their baby's diagnosis and condition. Communication about their baby's progress and the treatment plan was therefore very important

> *'If you're in the dark that's when people would worry [….] as long as you, you know, either good or bad, what was going on and a lot of the things you ask them they don't know the answer because it's a time will tell kind of answer, but so long as someone tells you the time will tell sort of answer you've got an answer.* (ID25, mother, daughter with jejunal atresia)

There were often extended periods of uncertainty as doctors were not able to answer parents' questions. While these periods were a challenge to parents, there were examples of good, clear communication which really helped at a stressful, frightening time.

### Being kept up to date

Being kept up to date with their baby's care was important to parents, especially when plans changed. They valued the opportunity to ask questions repeatedly. ID25 said she was constantly asking questions but staff were very good at answering them all and reassuring her.

> [I] *probably annoyed the hell out of them, but wanted to know, OK, when he does this what is his next step, what has progressed to, how long will he be doing that and how long will it take and things.* (ID15, mother, son with CDH)

> *The nurses were good as well at feeding back what the doctors had said and also if you had any questions I felt comfortable saying look can you ask about this and let me know what they say, so.* (ID30, mother, son with gastroschisis)

### Communicating well

Although health professionals often had to deliver bad news, there were many examples of good practice in managing this well. These interactions were characterised by giving parents information in a clear, accessible manner, sometimes supported by drawing diagrams, but without patronising or 'dumbing down'. Parents appreciated doctors and nurses taking time to answer their questions, not making them feel that any question was a silly one.

> **M**: *And went through them with the consultant and actually again that was another thing that he very patiently sat*

*and went through every single one of those questions, however stupid they were, you know.* **F:** *How many times he'd heard them, no doubt.* **M:** *Yeah and it didn't.* **F:** *He wasn't fazed, he wasn't awkward he was just 'Okay, let's do it'.* (ID 43/44, mother and father, son with Hirschsprung's disease)

*I quite like things in a lot of detail so that when doctors are talking to you sometimes they can do stupid talk because they don't know how much you know medically, but I like to hear all of it and I don't want broken down terms, I want medical terms because then I can go and look up what the medical terms are, see what it's all about.* (ID25, mother, daughter with jejunal atresia)

### Conveying expertise

When infants were in hospital for long periods, relationships of mutual respect between parents and healthcare professionals were hugely valued. As one mother explained, "*you are handing over the most precious thing to this person who is going to put them under anaesthetic and disappear into an operating theatre with them and you hope their gonna come out the other side. So you have to have that level of assurance.*" She and her husband were reassured by their surgeon's 'confidence not arrogance'. Others talked about the importance of trust and feeling as though their baby was in expert hands, cared for by clinicians who were knowledgeable about their condition.

*They've got this under control, they really know what they are doing…… In a way it's silly thinking that now because of course they know what they're doing these are highly trained professionals but in that moment that's what you want to hear cos you just want to, you're handing over your child so you want someone to be right on it don't you and he was."* (ID26, father, daughter with inguinal hernia)

### Being listened to and feeling part of the team

While parents recognised that they relied on doctors' expertise and judgement in planning their baby's care, they really valued feeling involved in caring for their baby where they could—'*part of it, part of that team*'.

*They've been very understanding and patient and I think recognising as well our role as parents and, that we, you know, have opinions and experiences, and not devaluing what we've learnt from other places as well.* (ID01, mother, son with exomphalos)

*I would say 99.9% of the time we've been treated really well, like human beings, not just numbers, and that has made an enormous difference to the experience as a family.* (ID03, mother, daughter with exomphalos)

But communication did not always go so well and parents could find this very undermining, particularly at such a stressful and worrying time. One parent (ID37) felt that the doctors looking after her daughter did not respect her wish to breast feed, and had an attitude that parents were part of the problem rather than part of the solution, which frustrated her.

Unlike parent ID01, one mother (ID19) felt that doctors were very dismissive of the research and expert patient knowledge she had gathered from online support groups.

*So yeah the surgeons need to learn how to not go 'Oh the internet' every time you suggest that someone has [laughs] mentioned this to you [laughs] because it's not just Googling symptoms and getting an answer it's asking a forum of parents who've been through exactly what you're going through. It's like sitting in a room with 500 parents whose kids have got the same experience and had the same problem who'll all have widely different experiences of it and the surgeon just going 'I'm not listening to any of you lot,' what surgeon would stand in a room of parents and do that but because its's on the internet it seems to be acceptable to go 'Tut, Facebook.'* (ID19, mother, daughter with exomphalos)

### Managing information and information overload

Understanding the diagnosis and possible journey ahead can be overwhelming for parents. Parents often felt powerless, both during pregnancy and once their baby had been born. Information—about the condition and procedures but also other parent experiences—was crucial in helping parents come to terms with the diagnosis. Seeking out information played a central role, helping parents feel they were actively doing something for their baby and family. ID01's son had exomphalos. Seeking information felt like the only thing she and her husband could do. "*We were powerless to change anything other than lots of information.*"

However, there were also dangers of information overload. Parents could find it hard to take on board a great deal of complex information at once, and needed time to understand and process what was happening to their baby. ID22's son was diagnosed with Hirschsprung's disease. She was given an information sheet and some medical alert cards so they could identify any symptoms of things going wrong.

*They gave us enough information but not too much because I think they obviously realised that, you know, it's early days but they had to highlight the seriousness of it and that, know if anything if he was showing any signs of anything you're to bring him back basically.* (ID22, mother, son with Hirschsprung's disease)

On the whole, parents said the information given by hospitals was not sufficient and they supplemented it with their own background reading and research. Pointing parents in the direction of trusted sources of information would be a key easy action to help them.

### Encouraging parents to seek out others

Finding support from other parents who had been through similar experiences (either online or face to face) was described as a crucial factor in helping parents cope and understand more about their baby's diagnosis.

*Find parents, my absolute number one thing.* (ID01, mother, son with exomphalos)

*That was my number one support during that time with all these other mums who'd gone through it.*(ID02, mother, daughter with gastroschisis).

Staff can encourage and facilitate this through introducing parents to each other, or suggesting online support groups. One unit had established a popular and thriving Facebook support group for parents of babies with Hirschsprung's disease.

*It just makes it a bit real, you know, that you aren't the only one.* (ID22, mother, son with Hirschsprung's disease)

### Helping parents find a role
Parents often described how 'useless' or 'helpless' they felt not being able to do anything for their baby while in hospital. Yet when they went home they were going to become experts in their baby's care. Helping them bridge this gap was an area where staff could make a real impact.

*I think you're so wrapped up in the medical needs of your baby and hoping that they're going to be alive at the end of it that I just…hadn't anticipated that the little things would be the things that matter, it would be the 'mummy' things that had been so easy to do with my other children, they would be the things that would matter that I missed out on.* (ID19, mother, daughter with exomphalos).

In the early days, spending time at their baby's side was all many parents felt they could do. They looked for small ways they could regain a sense of control and get involved in their baby's care. While it was not always practical in critical care environments, parents appreciated helping with feeding, bathing, singing and talking to their baby where possible. After 3weeks, ID02 was able to hold her daughter for the first time, '*it was like Christmas*'. Supporting mothers to either breast feed or, more often, express milk for their baby, was another way of parents bonding and being involved in their baby's care. Some parents described how they were supported and encouraged to become 'experts' in aspects of their baby's daily care—changing dressings or stoma bags. One father (ID05) said he became the 'dressing king', after a rocky start. Another (ID03) became an expert at dressing her daughter's exomphalos,

*treating and dressing it was still a bit of a skill. It got easier and easier and it's a real shame. I'm now one of the finest exomphalos wrappers in [county] and it's not a skill that's really going to be used that much, is it?*

Activities like these were key to bridging the gap between hospital and home.

### Preparing to go home
After weeks or months in hospital, being able to bring their baby home was a huge step for many parents. While a positive sign that their baby was on the mend, it could also be daunting to leave the 'safety net' of hospital and clinical expertise. Some parents described handover or normalisation programmes as very helpful in preparing them technically and emotionally for caring for their infant at home. Parents were trained in some of the skills they would need (wound management, stoma care, first aid, resuscitation) and given the opportunity to 'room in' with their baby for a few nights before going home.

*You're totally encouraged to do as much care for your baby as you can. It's your baby, they're there to support you and to medically [um] jump in if, if need be, but don't expect that they're there to feed and care for your baby because that's your job.* (ID34, mother, son with gastroschisis)

### Open access policy
Accessing hospitals and specialists after their baby had been sent home could be challenging for parents, especially if home was a distance from the specialist surgical centre. When problems came up, as they often did, it was hard to know where to take their baby. So, being given open or emergency access back to the team who had looked after their baby, without having to go through the Accident & Emergency Departmentfirst, was really valued by parents. ID38's son had complex needs after his surgery for oesophageal atresia and tracheo-oesophageal fistula. She said having open access to the ward where they can turn up with their baby and see doctors who know their baby was so important. "*I know that won't last forever but right now the open access is a Godsend because it means we get seen really quickly.*"

### DISCUSSION
This study adds valuable insights into the support and information needs of parents when their baby requires early abdominal surgery. These surgical journeys are often a long haul for baby and parents. Our study highlights the importance of clear and ongoing communication between parents and the often large, multidisciplinary teams caring for their infant and points to immediate actions that can improve parent experiences and outcomes. Parents value being listened to, encouraged to be a part of the care team and supported in finding a role for themselves. They need information from the professionals caring for their infant but also recommendations of good sources of information to supplement their knowledge, while being mindful of information overload. For some, encouraging them to seek support from other parents is of great value.

Previous work exploring parents' experiences of having an infant in NICU or surgery has highlighted the importance of good communication and emotional support, giving parents support at handover for surgery and pointed to the impact of scanty and infrequent information giving.[7 32] Parents experiences are affected by their baby's diagnosis and care, and the ways in which services are organised to support families to navigate the systems

and maintain a new 'second campus' as they look after their baby and continue their lives outside the hospital.[33]

Involving and supporting parents are increasingly recognised as an important component of providing holistic paediatric care and can help with the uncertainty of parenting an infant during critical illness. Parent experiences of NICU point to the vital role that neonatal nurses have in engaging with, in particular, mothers and the importance of that relationship.[33] Aagaard and Hall[5] discussed the importance of trying to strengthen maternal competence, and suggested parent–nurse chat as a communication strategy. Our findings provide examples of the ways in which encouraging parents to be involved with their infant can be achieved.

Others have described the role of the expert parent[33] and home monitoring programmes after cardiac surgery.[17 34] Studies have indicated that parents are not adequately prepared for discharge and are not well equipped to recognise deterioration in their child. The positive examples of being prepared for discharge and supported in becoming the primary carer offer insights into how parents might be supported to manage this transition.

Bourdieu's theoretical concepts of habitus and capital are helpful in exploring and unpacking the experiences of parents in these highly technical environments and the value they place on communication, finding their own role and seeking out other parents. They are on a journey from being newcomers in an environment (neonatal/surgical unit) where they are unfamiliar with the socialised norms. They have little capital and no idea how to 'play the game' (what Bourdieu termed the logic of practice) and negotiate the field. There is initially an unequal balance of power. Thus, examples of parents feeling dismissed (or part of the problem) amplify this imbalance. Feeling included in the team caring for their infant and having their expertise (in changing dressings, feeding, etc) supported and celebrated can be a powerful tool in addressing this imbalance. Over time, assisted by good communication from the health professionals and the networks of support and information provided by other parents, parents develop confidence and competency–which Bourdieu might have interpreted as social and cultural capital. Their social capital is extended through the social networks they develop inside and outside the hospital environment, online and in the real world, relating to their new role in caring for their infant. Their cultural capital is extended through the knowledge, expertise and skills they develop to care for their infant. By the time they take their infants home, these parents are well on the way to becoming technical experts in their baby's illness and care.

### Strengths and limitations

This is a UK-based study that sought to explore the experiences of parents who have an infant requiring abdominal surgery in the first year of life. Interview studies in this field are rare and our study provides rich insights into the emotional and practical impacts of these experiences. While the interviews could not cover all possible conditions, they provide insights that may be generalisable across many different surgical procedures/medical conditions. However, there are several limitations worth considering. We interviewed parents at different lengths of time after their surgery, as we sought to gain insights into the long-term as well as short-term impacts of these experiences. However, this variation could have influenced parents' recall of events. Parents were asked to describe an intense and often protracted period of their lives in one interview, so it is perhaps inevitable that some details were omitted or compressed. There was also a variety in interviews, with some parents choosing to be interviewed alone, and others in couples. This inevitably means that some interviews are coproduced while others are not. There was little conflict in evidence during interviews although there were inevitable differences of opinion and recall.

## CONCLUSION

As babies recover, the potential for parental expertise grows. Our study highlights the vital role of information, both about the condition and about the infant's progress. Healthcare staff could discuss or direct parents to information sources, seek to empower parents through involving them with their infant's care where possible and link them with other parents.

**Contributors** The study was conceived by MK. All interviews were collected by LH. Analysis was undertaken by LH and LL, with input from MK and A-ML. LH wrote the article, with input from the other authors. The authors thank the members of our parent advisory (PPI) group for their contributions throughout the study.

**Funding** This study was funded through a National Institute for Health Research (NIHR) Professorship award to MK (NIHR-RP-011-032) and supported by National Institute for Health Research Biomedical Research Centre, Oxford, grant BRC-1215-20008 to the Oxford University Hospitals NHS Foundation Trust and the University of Oxford.

**Disclaimer** The views expressed are those of the author(s) and not necessarily those of the NHS, the NIHR or the Department of Health.

**Competing interests** None declared.

**Patient consent** Parents consented to interviews and approved transcripts for analysis and publication.

**Ethics approval** Berkshire Ethics Committee, 09/H0505/66.

**Provenance and peer review** Not commissioned; externally peer reviewed.

**Data sharing statement** Participants were invited to review their transcript and mark any sections that they did not want used before transferring copyright to the University of Oxford for use in research, teaching, publications and broadcasting. These carefully anonymised transcripts form part of a University of Oxford archive which is available to other bona fide research teams for secondary analysis. All authors had access to all of the data in the study and take responsibility for the integrity of the data and the accuracy of the data analysis.

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
