## [Reviewer comments · BMJ Open]

ARTICLE DETAILS

TITLE (PROVISIONAL)	What can make things better for parents when babies need abdominal surgery in their first year of life? A qualitative interview study in the United Kingdom
AUTHORS	Hinton, Lisa; Locock, Louise; Long, Anna-May; Knight, Marian

VERSION 1 – REVIEW

REVIEWER	Daniel Perry University of Liverpool, UK
REVIEW RETURNED	08-Dec-2017

GENERAL COMMENTS	I enjoyed this paper, and as a children's orthopaedic surgeon I thought that this was very enlightening in places. I am not an expert in qualitative methodology (nor am I clear about the reporting guidelines), though as an academic clinician in this area I thought that it was very helpful, and will change my practice in terms of encouraging the use of social media.
---

REVIEWER	Dr Kate Oulton Great Ormond Street Hospital for Children, UK
REVIEW RETURNED	21-Dec-2017

GENERAL COMMENTS	This is an interesting study, focusing on a potentially challenging time for parents - that of their infants having surgery. The paper is largely well written and easy to follow but a lack of detail throughout, leaves multiple questions unanswered, and I feel does not do justice to the data likely to have been collected in a study of this type and size. Detailed comments are provided below, which if addressed will strengthen the quality of the paper and provide clarity to the reader, from which they can then draw conclusions about the relevance of the findings to their work. Introduction The authors point to the lack of research into conditions that require early surgery (p4, line 40), suggesting a need to supplement the evidence base by drawing on research into experiences of congenital heart disease and surgery - it is unclear why the latter is considered supplemental and not part of the evidence base that exists. Aim The authors use a number of terms interchangeably throughout the paper, which make the primary aim unclear - is this about infants who require surgery in the first year of life (p6, line 33) or who undergo neonatal surgery (page 2, line 26) – this needs clarifying.
--

	A further aim of this study is to identify the questions and problems that matter to parents during and after their pregnancy and the long-term impact on parents and families (p6, line 19). The results described focus on the time after surgery and make little reference to pregnancy or long-term impacts. It is suggested that further data is added to address this aim or the paper focuses solely on the parental experiences of their infant undergoing surgery. Participants More information about the study participants is required and should be described under a dedicated heading, not within the methods section. The table is useful, but a few sentences summarizing the participants would help ease of reading. The paper is lacking important information about the age at which infants had their surgery, whether the condition was congenital or not, whether the surgery was major or minor, those that required intensive care - these factors are referenced in the background section as relevant to the parental experience so should be drawn out in the discussion. Socio-demographic information about the parents involved, including the number of mothers and fathers is also needed. Little if any reference is made to the paternal experience throughout the paper, despite the study seeming to include 11 fathers – this needs addressing. The study sample also includes parents whose child had surgery ‘very recently’ to 25 years previously. This description is too vague to be meaningful to the reader – whilst the table includes the child’s age when the parent was interviewed this does not inform the reader about the child’s age at the point of surgery – which may be an important factor in the results. The challenge of managing data spanning such a large time frame is also not addressed nor is the reason for doing this. Parental recall of the situation, for example, is not discussed in the limitations, nor the changing attitudes or medical practices that may have changed over this time. Methods Please clarify whether 42 (page 6, line 31) or 44 (page 2, line 24) parents were interviewed. The methods section requires more detail. Both individual and paired interviews are reported but no information is given about how the latter were managed or the dynamics that occurred and the impact this had on the data collected. The method of ‘narrative’ interviews is not justified or described nor is the process of videoing interviews – why was this method used, what did it add, how was the data managed? The ethical issues associated with videoing are not mentioned. The data analysis section is poorly described. What process was used for analyzing video data for example and why was modified grounded theory used for interview transcripts and not narrative analysis? Please also state how long the interviews lasted. There are no papers on videoing or narrative analysis in the reference list – please address this. Results and discussion Some interesting results are presented about the parental experience. The authors describe the production of rich data from a large sample of parents, as would be expected with the use of narrative interviews, however, very little data is presented to support the themes described. For example the “deep and lasting distress and anxiety some parents feel” (Page3, line 14) is not sufficiently conveyed.
--	--

	Some of the quotes are included in Box 1, but not others and the basis for choosing when to do either one is not clear. For example, some themes have more than one quote in the text, whilst others have none – I would suggest taking a more consistent approach to the presentation of the findings - giving each theme equal attention. Some of the themes overlap closely and their distinctiveness is not always clear – for example, the value parents feel in being involved in their baby's care, under the theme – 'being listened to and feeling part of the team' links closely to 'helping parents find a role' – further data would help illustrate the distinction between themes or the relationship between them. Many of the quotes in the box are interesting and warrant further analysis and discussion – for example, "not devaluing what we've learnt", "we've been treated really well, like human beings", "I probably annoyed the hell out of them" – how do these link to Bourdieu's theory of habitus and capital – if this framework is to be used then I would suggest greater depth of analysis and discussion is needed. Quotations Please do not refer to parents by their ID number in the text (see for example, p9, line 28) and in the quotes that are taken from parent dyad interviews make it clear whether it is the mother or father who is being quoted. Overall, this paper has potential for publication and drawing on Bourdieu's work is interesting but at present the lack of detail and consistency in the sampling and methods sections leaves too many questions unanswered and the results section is underdeveloped. I couldn't help feeling disappointed that I did not get to read more about the 'stories' these parents told and perhaps a journal with a longer word count would do more justice to the data and methods utilised.
--	--

VERSION 1 – AUTHOR RESPONSE

Reviewer: 1

- I enjoyed this paper, and as a children's orthopaedic surgeon I thought that this was very enlightening in places. I am not an expert in qualitative methodology (nor am I clear about the reporting guidelines), though as an academic clinician in this area I thought that it was very helpful, and will change my practice in terms of encouraging the use of social media. We are delighted that Reviewer 1 enjoyed the paper and found it enlightening.

Reviewer: 2

Reviewer 2 enjoyed our paper but made detailed comments, inviting us to clarify and strengthen the paper. We have addressed them as follows:

- Introduction

The authors point to the lack of research into conditions that require early surgery (p4, line 40), suggesting a need to supplement the evidence base by drawing on research into experiences of congenital heart disease and surgery - it is unclear why the latter is considered supplemental and not part of the evidence base that exists.

We have added details (p.5) about the wider study that this interview study was related to. Its focus was on abdominal surgery. We hope that this clarification will make clear why we felt that insights into cardiac surgery were therefore a different evidence base.

- Aim

The authors use a number of terms interchangeably throughout the paper, which make the primary aim unclear - is this about infants who require surgery in the first year of life (p6, line 33) or who undergo neonatal surgery (page 2, line 26) – this needs clarifying.

We have replaced neonatal with “early surgery” throughout the paper.

- A further aim of this study is to identify the questions and problems that matter to parents during and after their pregnancy and the long-term impact on parents and families (p6, line 19). The results described focus on the time after surgery and make little reference to pregnancy or long-term impacts. It is suggested that further data is added to address this aim or the paper focuses solely on the parental experiences of their infant undergoing surgery.

We have clarified at the end of the Background section that this paper reports only on the period immediately before and after surgery (p.7)

- Participants

More information about the study participants is required and should be described under a dedicated heading, not within the methods section. The table is useful, but a few sentences summarizing the participants would help ease of reading. The paper is lacking important information about the age at which infants had their surgery, whether the condition was congenital or not, whether the surgery was major or minor, those that required intensive care - these factors are referenced in the background section as relevant to the parental experience so should be drawn out in the discussion.

Socio-demographic information about the parents involved, including the number of mothers and fathers is also needed. Little if any reference is made to the paternal experience throughout the paper, despite the study seeming to include 11 fathers – this needs addressing. The study sample also includes parents whose child had surgery ‘very recently’ to 25 years previously. This description is too vague to be meaningful to the reader – whilst the table includes the child’s age when the parent was interviewed this does not inform the reader about the child’s age at the point of surgery – which may be an important factor in the results. The challenge of managing data spanning such a large time frame is also not addressed nor is the reason for doing this. Parental recall of the situation, for example, is not discussed in the limitations, nor the changing attitudes or medical practices that may have changed over this time.

We have written a distinct ‘Participants’ section with the details suggested, and added details to the Table. We have expanded the Strengths and Limitations section in the Discussion (p.20) to include reflection on the limitations of parental recall.

- Methods

Please clarify whether 42 (page 6, line 31) or 44 (page 2, line 24) parents were interviewed. The methods section requires more detail. Both individual and paired interviews are reported but no information is given about how the latter were managed or the dynamics that occurred and the impact this had on the data collected. The method of ‘narrative’ interviews is not justified or described nor is the process of videoing interviews – why was this method used, what did it add, how was the data managed? The ethical issues associated with videoing are not mentioned. The data analysis section is poorly described. What process was used for analyzing video data for example and why was modified grounded theory used for interview transcripts and not narrative analysis? Please also state how long the interviews lasted. There are no papers on videoing or narrative analysis in the reference list – please address this.

We have clarified the interview numbers and provided further details in the Methods section to address the issue of paired and individual interviews. We have given more detail about the interviews, which we conducted using a semi-structured narrative approach. They were video recorded, where consent was given, to facilitate publication of the patient experience web resource Healthtalk.org. Ethics approval was obtained for video or audio recording of the interviews, but there was no obligation for interviews to be video recorded. All material quoted in articles, or published on the Healthtalk website, have been approved and copyrighted by participants. We have extended our

description of our analysis and given details of the length of the interviews. Further references have been added.

- Results and discussion

Some interesting results are presented about the parental experience. The authors describe the production of rich data from a large sample of parents, as would be expected with the use of narrative interviews, however, very little data is presented to support the themes described. For example the “deep and lasting distress and anxiety some parents feel” (Page3, line 14) is not sufficiently conveyed.

The Results section has been revised, in particular to include the quotes in the main body of the results, so they can more closely illuminate the findings being presented. We have added quotes (on p.10) to strengthen our claim that parents felt lasting distress and anxiety.

- Some of the quotes are included in Box 1, but not others and the basis for choosing when to do either one is not clear. For example, some themes have more than one quote in the text, whilst others have none – I would suggest taking a more consistent approach to the presentation of the findings - giving each theme equal attention. Some of the themes overlap closely and their distinctiveness is not always clear – for example, the value parents feel in being involved in their baby’s care, under the theme – ‘being listened to and feeling part of the team’ links closely to ‘helping parents find a role’ – further data would help illustrate the distinction between themes or the relationship between them. Many of the quotes in the box are interesting and warrant further analysis and discussion – for example, “not devaluing what we’ve learnt”, “we’ve been treated really well, like human beings”, “I probably annoyed the hell out of them” – how do these link to Bourdieu’s theory of habitus and capital – if this framework is to be used then I would suggest greater depth of analysis and discussion is needed.

We hope that by moving quotes to the main body of the article, and our expanded discussion, in particular in relation to the theoretical lens of Bourdieu, addresses the reviewer’s concerns outlined here.

Quotations

Please do not refer to parents by their ID number in the text (see for example, p9, line 28) and in the quotes that are taken from parent dyad interviews make it clear whether it is the mother or father who is being quoted.

We have left the participants as ID numbers in this version, but if the editor is willing, we would be very happy to replace throughout with pseudonym names. We have clarified in the dyad interviews, who is speaking.

VERSION 2 – REVIEW

REVIEWER	Dr Kate Oulton Great Ormond Street Hospital, UK
REVIEW RETURNED	02-Mar-2018

GENERAL COMMENTS	This paper is much improved as a result of the revisions made. There are some minor revisions I believe are necessary before publication.  1. Add limitations to the article summary at the beginning of the paper 2. Clarify in the main body text that this paper is about abdominal surgery, not all surgery - this is clearly in the title but not in the abstract nor the study aim/main body of the paper. 3. Give further thought to the type of interviews conducted - I do not feel in-depth, semi-structured narrative interviews is appropriate -
--

	see the following paper: METHODOLOGICAL ISSUES IN SOCIAL HEALTH AND DIABETES RESEARCH Year : 2013 Volume : 1 Issue : 2 Page : 56-59 Three types of interviews: Qualitative research methods in social health Heather L Stuckey Narrative interviews are led by participants and whilst you started with an open-ended question, the remainder of the interview appears researcher led. I would suggest in-depth interviews with prompts is a more appropriate description taking into account their extended length, or perhaps semi-structured. It might be helpful to upload the interview schedule for readers. 4. Did anything come up from fathers data that was different from mothers, or was there any conflict during interviews conducted with both parents - either way it would be helpful to know. 5. The paper requires a final read through to check consistency in the way quotes are presented, use of acronyms and sentence structure in a few places.
--	---

VERSION 2 – AUTHOR RESPONSE

Dear Editors,

Thank you very much for your further comments on our paper and giving us an opportunity to strengthen and clarify our findings. We have responded to the suggestions from Reviewer 2 as follows and provide a track changes and clean copy of the revised manuscript.

1. We have added limitations to the article summary at the beginning of the paper.
 2. We have clarified in the abstract and main body of the article that the paper refers to abdominal surgery, not all surgery.
 3. We welcome the opportunity to further clarify the interview approach. We have provided further detail in the body of the text and amended the abstract to just use 'in depth'. The interview was in two parts. In the first part, we invited parents to tell us their story in their own words (the unstructured narrative) and this part of the interview was often lengthy. This was followed a semi-structured component of the interview using prompts to follow up issues raised in the narrative and to explore themes suggested by the literature and advisory panel.
 4. Reviewer 2 raises a question about differences between the father's and mother's accounts. An analysis of the differences in fathers and mothers accounts is beyond the scope of this paper, but there was no striking conflict. We have added a sentence to this effect in the Strengths and limitations section.
- The author also provided a marked copy with additional comments. Please contact the publisher for full details.

VERSION 3 – REVIEW

REVIEWER	Dr Kate Oulton Centre for Outcomes and Experience Research in Children's Health, Illness and Disability (ORCHID), Great Ormond Street Hospital, UK
-----------------	---

REVIEW RETURNED	02-May-2018
-------------

GENERAL COMMENTS	All issues raised have been adequately addressed
--